# Nanoporous Layers and the Peculiarities of Their Local Formation on a Silicon Wafer

**Vitali Vasil'evich Starkov [1], Ekaterina Alexanrovna Gosteva [2,3,\*], Dmitry Dmitry Zherebtsov [4,5],
Maxim Vladimirovich Chichkov [2] and Nikita Valerievich Alexandrov [2]**

[1] Institute of Microelectronics Technology and High Purity Materials, Russian Academy of Sciences,
Ossipyan Str., 142432 Chernogolovka, Russia; starka@iptm.ru

[2] Department of the Material Science of Semiconductors and Dielectrics, National University of Science and
Technology MISiS, 4 Leninskiy Prospekt, 119049 Moscow, Russia; maxim.chichkov@gmail.com (M.V.C.);
adelsandrov@mail.ru (N.V.A.)

[3] Academy of Engineering, Peoples Friendship University of Russia (RUDN University),
6 Miklukho-Maklaya Str., 117198 Moscow, Russia

[4] Laboratory for Microparticle Analysis, 117218 Moscow, Russia; Dmitry_Zherebtsov@bk.ru

[5] Center for Composite Materials, National University of Science and Technology "MISiS",
119049 Moscow, Russia

\* Correspondence: gos-3@mail.ru

**Abstract:** This review presents the results of the local formation of nanostructured porous silicon (NPSi) on the surface of silicon wafers by anodic etching using a durite intermediate ring. The morphological and crystallographic features of NPSi structures formed on n- and p-type silicon with low and relatively high resistivity have also been investigated. The proposed scheme allows one to experiment with biological objects (for example, stem cells, neurons, and other objects) in a locally formed porous structure located in close proximity to the electronic periphery of sensor devices on a silicon wafer.

**Keywords:** porous nanoparticles; nanostructures; synthesis strategies; nanostructured surface

## 1. Introduction

The nanoscale textured porous silicon (NPSi) is gaining great interest for applications in various fields of science and technology. The main characteristic of NPSi which prevails in most cases of practical applications is a very significant specific surface area (up to 1000 m$^2$/g). The first research materials on the storage of hydrogen in the volume of NPSi have already been published [1–4]. Further development of work in this direction will make it possible to integrate such a hydrogen source with its consumers on a single silicon wafer. In this regard, it is important to carry out work using the porous silicon (PSi) in the structures of micro-fuel cells based on silicon, for which hydrogen is a fuel [5]. Works on the formation of various membrane and electrode structures based on NPSi are of considerable interest [6,7].

Another area of application of NPSi is the modern development of neuromorphic engineering research. Tests are being successfully carried out to develop the architecture of the surface of a microelectronic sensor, on which biological neural networks could be cultivated [8,9]. While creating neuroprocessors, such sensors must have high-tech electronic interfaces to implement communication with neural networks [10–12]. Considering that cellular structures, for example, based on pluripotent or neuronal stem cells, are able to transform (differentiate) into neurons (see, for example, in [10]), studies were carried out on the cytotoxicity of the nanostructured surface of silicon wafers to mammalian cells [13]. Samples of pure silicon, as well as NPSi, are not cytotoxic. At the same time, there is a tendency towards an increase in viable cells incubated on nanoporous samples compared to control structures (107% of viable cells for NPSi samples). This may indicate that the

created developed porous surface favors cell growth. (this may indicate that the resulting porous developed surface favors cell growth).

This review presents the results of the local formation of NPSi on the surface of silicon wafers by anodic etching using a durite intermediate ring. The morphological and crystallographic features of NPSi structures formed on n- and p-type silicon with low and relatively high resistivity have also been investigated.

## 2. Materials and Methods

### 2.1. Materials and Reagents

Silicon wafers of n- and p-types, heavily doped with arsenic and boron, were used for the formation of NPSi structures. The resistivity did not exceed 0.002 $\Omega$ cm; the orientation of the surface of the plates was (100). NPSi structures were made by anodic etching in a HF:$C_2H_5OH$ = 1:1 solution with the addition of a $10^{-3}$ M concentration of cetyltrimethylammonium chloride (CTAC:$CH_3(CH_2)_{15}N(CH_3)_3Cl$). Etching was carried out at a constant current density J = 80 mA/cm$^2$ at room temperature. The formation of NPSi structures on n-type silicon wafers was carried out without additional illumination in the etching area.

To obtain gradient porous layers with a variable morphology (GPS-var structure), p-type silicon wafers with a higher resistivity $\varrho_v$ = (10–1000) $\Omega$ cm were used [14]. The formation of GPS-var structure was conducted by anodic etching in solution HF:$C_3H_8O$(ISO) + $10^{-3}$ M CTAC at a constant etching current J = 10 mA/cm$^2$ at room temperature.

### 2.2. Obtaining Porous Layers on the Surface of Silicon Wafers

We used silicon wafers of 100 mm in diameter. Before carrying out the anodic etching, processes on the front side of the plate formed an oxide layer 0.2–0.3 mm thick. On the reverse side of the plate, a contact layer of aluminum by vacuum deposition was formed. On the oxide-coated plate (numbered 2 in Figure 1a,b), a durite intermediate ring (numbered 3 in Figure 1) was located in the place required for etching.

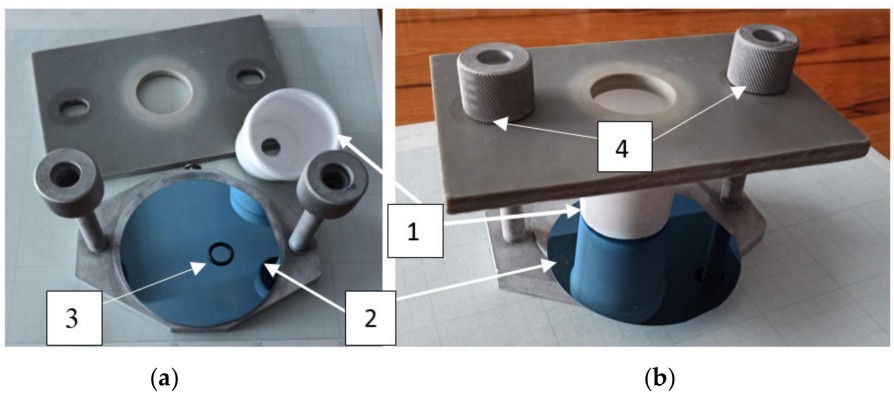

| (a) | (b) |

**Figure 1.** Device for local anodic etching. (**a**) disassembled cassette: 3—durite intermediate ring, 2—Si wafer (front (working) surface), (**b**) Cassette "assembled": 1—Fluoroplastic container for electrolyte, 4—fasteners.

An electrolyte-resistant fluoroplastic container (1), with a matching ring (3) hole in the bottom, is positioned on the ring and secured with mechanical fasteners (Figure 1b).

At the first stage, we carry out a local removal of the oxide film in the circular area with the diameter of the intermediate ring. Oxide etching is carried out using standard solutions based on hydrofluoric acid (for example—HF:$H_2O$ = 1:10).

After removing the oxide from the silicon surface inside the ring and drying with a stream of warm air, one or another mode of anodic etching of the exposed silicon region can be carried out (without additional disassembly of the cassette). Thus, the proposed etching process allows the formation of a porous structure locally, at the same time, minimally

affecting on the region of the silicon wafer, which was not selected for anodic etching. If more accurate local topological formation of the porous region is required, more complex photolithography processes can be used [15].

### 2.3. Research Equipment

The structure of samples and their elemental composition were examined by scanning electron microscope (SEM) Tescan LYRA 3 equipped with X-Max 80 EDX detector for X-ray microanalysis by Oxford Instruments with Aztec control system. SEM images were collected using an SE detector with an accelerating voltage of 10 kV. Overall, 200,000 pulses were accumulated for each elemental composition measurement.

The samples were examined by force microscopy in a tapping mode, cantilever NSG10, tip curvature radius 10–15 nm, at a scanning probe nanolaboratory (NT-MDT, Moscow, Russia).

The studies were carried out on a JEM 2100 transmission electron microscope (JEOL, Tokyo, Japan) at an accelerating voltage of 200 kV.

### 3. Results

#### 3.1. Nanostructured Layers of GPS-var Structures

Formation features and morphological characteristics of gradient-porous silicon structure with variable pore morphology were set out in [14,16]. For GPS-var, the structure is characterized by the presence of a nanoporous layer on the surface. Depending on the etching modes and, first of all, on the resistivity of silicon wafers, NPSi is observed at different depths of the porous layer. In the same etching modes for silicon wafers with resistance $\varrho_v = 10$ Ω cm or $\varrho_v = 1000$ Ω cm, the NPSi layer, to some extent, fills the volume of macropores. For silicon with lower resistance ($\varrho_v = 10$ Ω cm), NPSi is located at the entire depth of the pores (~31μm). In porous structures formed on silicon with higher resistance, NPSi is observed mainly in the near-surface macroporous layer, with a thickness 1–3 μm [14]. As noted in [6,14,16], the formation of both the surface nanostructure of the amorphous silicon layer and the complete or partial filling of the volume of the silicon skeleton are associated with the disproportionation reaction (precipitation of silicon atoms from the electrolyte [chemical reactions are presented, for example, in 14]). This etching process is carried out at a given anode current density and electrolyte composition.

Figure 2 shows the results of TEM and AFM analyses of the surface NPSi layer GPS-var structure. Based on the results of these analyses, it can be concluded that the nanoporous layer forms a highly developed porous surface with a pore diameter of (20–50) Å; the height difference (along the z axis) does not exceed Rms initial = 0.230 nm. For the initial surface of the silicon wafer, this value was Rms = 0.077 nm. The electron diffraction pattern (Figure 2b) indicates that the structure of the surface NPSi layer is amorphous (α-Si).

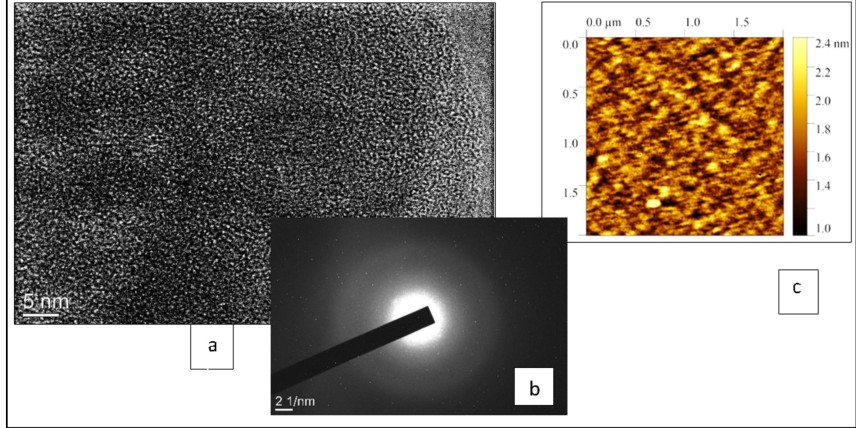

**Figure 2.** TEM and AFM analysis of superficial nanoscale GPS-var structure: (**a**) TEM image of the surface nanolayer; (**b**) electron diffraction pattern of the surface nanolayer. (**c**) AFM image of a fragment of the surface of the GPSi-var structure.

As noted in [6,14,16], the formation of the surface nanostructure of the amorphous silicon layer, as well as the complete or partial filling of the volume of the silicon skeleton, is associated with the disproportionation reaction (deposition of silicon atoms on the surface). This process proceeds at the given anodic etching current densities and electrolyte compositions to a greater or lesser extent, depending on the specific resistance of silicon.

### 3.2. Nanostructured Layers Based on Heavily Doped Silicon

Pore formation in silicon during anodic etching is related to anodic polarization in aqueous HF solution and depends on the electrode potential and HF concentration. Various mechanisms of the formation of porous layers are described in the literature [17–21]. Figures 3 and 4 show the results of the formation of nanoporous layers on silicon wafers of n-type and p-type conductivity, with the same resistivity $\varrho_v$ = 0.002 $\Omega$ cm, orientation of the surface of the plates (100). Anodic etching was carried out in a solution 1:1 = HF:Ethanol ($C_2H_5OH$). Etching was carried out at room temperature without any additional illumination of the samples in the mode of constant etching current density j = 80 mA/cm$^2$ within 60 min. The etching depth for both samples was (155–160) μm.

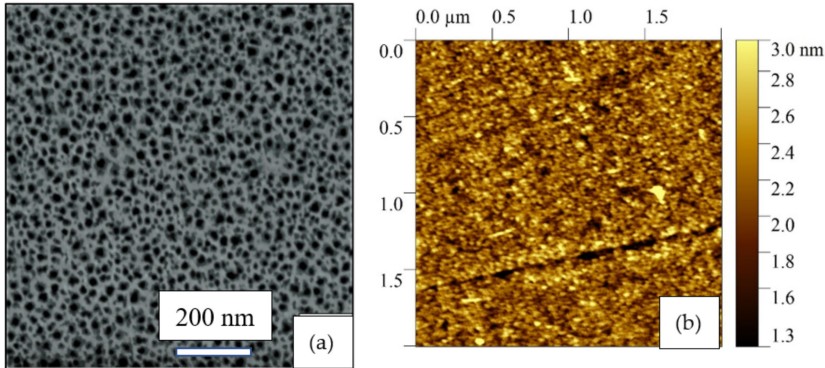

**Figure 3.** Surface of p-type silicon wafers with resistance $\varrho_v$ = 0.002 Ohm cm after NPSi formation. Images obtained with SEM-(**a**) and AFM-(**b**).

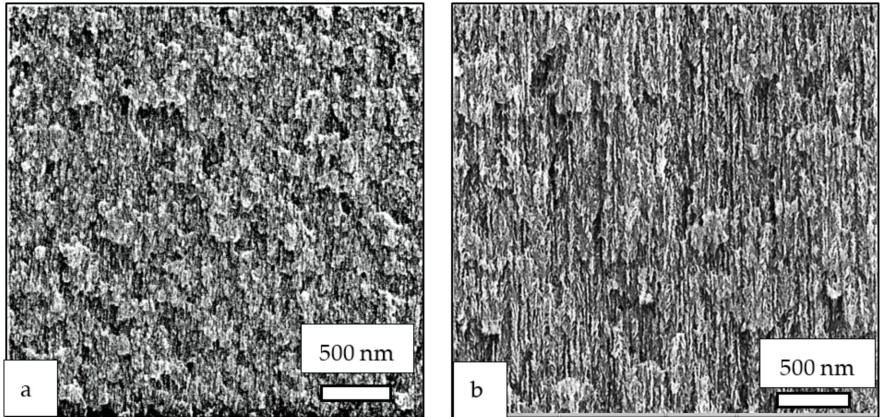

**Figure 4.** SEM images of cleavage fragments of porous layers of n-type (**a**) and p-type (**b**) silicon wafers after anodic etching.

The SEM analysis of NPSi structures of n-type and p-type Figure 4a,b shows that the pores for both structures are characterized by normal disposition to the etched surface. The pores are tubular. The diameter for the p-type is slightly larger than for the n-type pores. The porous layers have a crystalline structure. The porous layers are characterized by a surface, the typical SEM and AFM views of which are shown in Figure 3a,b. The roughness of the porous surface is Rms = 0.313 nm.

*3.3. The Edge Structures of the Porous Region*

A comparative analysis of the topological arrangement of the pores in the central part of the porous ring (Figure 3a) and at the edge of the porous region at a distance of about 1 mm from the intermediate ring (Figure 5a) indicates a significant difference in the mechanisms of pore nucleation in these regions. In the central part of the porous circle, this corresponds most closely to the quasi-hexagonal arrangement of the pores. The topological arrangement of pores (Figure 3a) corresponds to the defect–deformation (DD) mechanism of pore nucleation on the surface during anodic etching, which is described in detail in [22].

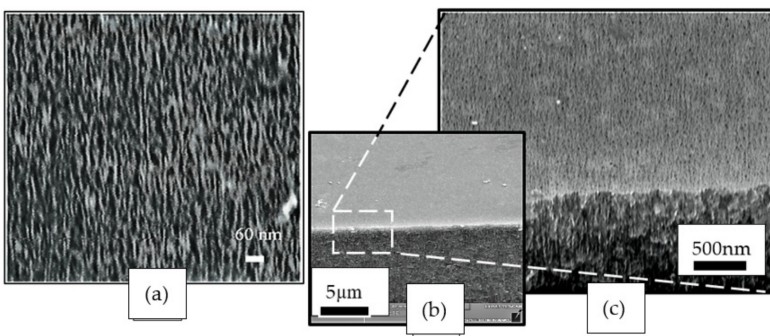

**Figure 5.** SEM image of the porous layer at the boundary of the porous layer with the outer sur-face of the silicon wafer: (**a**) surface image; (**b**) view of the angle of the chipping edge (5 μm); (**c**) view of the angle of the chipping edge (500 nm).

The arrangement of the pores in the immediate vicinity of the edge of the porous circle, in contrast to the central part, corresponds to the "slit" arrangement of the pores. Such a topological arrangement of pores is typical for structures in the process of anodic etching under conditions of compressive mechanical stresses [23]. A visual demonstration of the crevice nature of the pores is shown in Figure 5b,c. At the edge of the chipping, you can observe the spreading of the crevice pores from the surface to the volume on the chipping. Compressive stresses arise in the vicinity of the rubber ring, which is pressed against the plate using the fixing screws (No. 4 in Figure 1).

## 4. Conclusions

The processes of local formation of nanostructured regions on the surface of silicon wafers are considered. The presented results testify the possibility of practical realization of the process of localization of the porous region on the surface of silicon wafers, without any noticeable influence on the structure of the silicon wafer outside the porous region.

The porous structure is characterized by sufficient homogeneity over the entire area, except for the areas adjacent to the rubber spacer ring. This distance did not exceed 1 mm from the border of the ring and depends on the degree of clamping by the fixing screws.

The proposed scheme allows conducting experiments with biological objects (for example, stem cells, neurons, and other objects) in a locally formed porous structure located in the vicinity of the electronic periphery of sensor devices, which can be pre-formed on a silicon wafer.

**Author Contributions:** Conceptualization, V.V.S. and E.A.G.; methodology, V.V.S. and M.V.C.; software, D.D.Z.; validation, E.A.G. and V.V.S.; formal analysis, N.V.A.; investigation, D.D.Z., M.V.C. and N.V.A.; resources, V.V.S. and E.A.G.; data curation, N.V.A.; writing—original draft preparation, V.V.S.; writing—review and editing, V.V.S.; visualization, D.D.Z. and N.V.A.; supervision, V.V.S. project administration, E.A.G.; funding acquisition, E.A.G. and V.V.S. All authors have read and agreed to the published version of the manuscript.

**Funding:** This paper has been supported by the RUDN University Strategic Academic Leadership Program. The work was conducted according to the STATE TASK No. 075-00355-21-00.

**Institutional Review Board Statement:** Not applicable.

**Informed Consent Statement:** Not applicable.

**Acknowledgments:** TEM study was carried out on the equipment of the Center Collective Use "Materials Science and Metallurgy" with the financial support of the Russian Federation represented by the Ministry of Education and Science (agreement No. 075-15-2021-696).

**Conflicts of Interest:** The authors declare no conflict of interest.

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
