# Peer review of "Nanoporous Layers and the Peculiarities of Their Local Formation on a Silicon Wafer"

_processes, doi:10.3390/pr10010163_

Round 1

Reviewer 1 Report

The article details the etching conditions to obtain porous silicon with different morphologies. The results are supported with several physical characterizations (TEM/SEM & AFM) to help the reader to understand the morphology obtained for a given etching condition.

However, the paragraph named “2.2 Obtain porous layers on surface of silicon wafers” is hard to understand as it seems to me like a “usual” (I mean detailed in the literature for 30 years) way to produce porous silicon by electrochemical etching on monocrystalline Si. Maybe is there some novelty on the etching process given here that I mist or misunderstood.

Finally – in my opinion – hypotheses and references to the literature (among many others: XG. Zhang, J. Electrochem. Soc., 151, C69 (2004), V. Lehmann et al. Mater. Sci. Eng. B, 69-70, 11 (2000), RL. Smith, J. Appl. Phys. 71, R1 (1992), JN. Chazalviel et al. J. Electrochem. Soc., 149, C511 (2002)) are missing in the later part of the article, especially to relate the results presented here with the State of the Art in the domain.

As a conclusion, I recommend several (compulsory) minor revisions in the later part of the document before acceptance of the manuscript. The detail of the review is given below.

General recommendations:

L.16: The meaning of “durite” remains unclear to me. Is it related to the o-ring?

L25: references 1 & 2 could be completed by the article published by Lysenko in 2005 (V. Lysenko et al., Study of porous silicon nanostructures as hydrogen reservoirs, J. Phys. Chem.B, 109, 42, 19711 (2015)).

L25/27: Could the authors add at least one sentence detailing the mechanisms of hydrogen storage and use in NPSi.

L59/61: The meaning of this sentence is unclear to me.

L68: Was the oxide layer already grown on the wafers when purchased or was it intentionally grown? If so, what is the interest of starting the electrochemical etching from a SiO2 surface instead of “bare” Si surface?

L78/84: The explanations given in this paragraph remain unclear to me. To avoid any misunderstanding: the oxide layer is first removed in HF (1:10 vol. in H2O), and then (after rinsing) another HF-based solution was placed into the electrochemical cell. But, even though the opening between the first and the second HF solution immersion is not the same, there is no masking to protect a part of the substrate as HF (SiO2 etchant) is present in both cases. Unless the HF concentration in the second solution is highly diluted with H2O and ISO.

L112: Amorphous layer is mentioned without explanation (N.B. Explanation arrives L120/121). It is also interesting to mention that the amorphous nature of porous silicon is slightly different to some results obtained on similar substrate (i.e. Joo et al. J. Appl. Lett. 108, 153111 (2016)). These differences with the literature could be interesting to develop.

L113: Could you please detail (and reference) more on the disproportionation reaction mentioned L113.

L136: What is the reason for the change in additive in the electrolytic solution (from ISO (isopropyl alcohol?) to (I guess) ethanol)?

L142: It is interesting to notice that, as the pores are larger in p-type Si (as expected according to the literature), the etch rate should be significantly lower in p-type than n-type, which is not the case because L139, the porous layers on the two substrate have approximately the same thickness (155/160µm). Could you comment on it (maybe mentioning the Si dissolution valence in each case)?

L152: In my opinion, it is hard to compare the top view (fig.5a/b) and cross-section (fig. 5c/d) of two different etching conditions. Moreover, such differences could also come from an “edge-effect” (increase in local current density at the edges of the porous region).

Typo recommendations:

L.15: NPSi (I guess being nanoscale textured porous silicon) is not defined previously. It could be written in without abbreviation in the abstract in my opinion.

L61: I guess that GPS-var means “gradient porous silicon with variable morphology”. Please add the meaning of the abbreviation.

L114: The sentence starting with “This process proceeds at…” is not clear in my opinion. Would it be possible to rephrase it ?

L 131: “depends on” instead of “depending on”.

L136: The meaning of CTAC is already given L56 so, there no need to define it again here.

Author Response

On behalf of the authors, thank you for your interest and the work done. You helped make our work better.

Reviewer 2 Report

This contribution is acceptable. The corrections and queries are in file word attachment. The reference style must be improved. However, I consider that the number of self-citations is inappropriate. The review of different contributions should be increased for this manuscript.

Author Response

(The authors gave the same response as above.)

Reviewer 3 Report

1.The total manuscript should be written in detail, like the abstract should be deliver more information about the research.

2.In section 3.2 , the number of C2H5OH should subscript other than normal type. And there are similar errors also should be corrected carefully.

3.For the Figures 4, these figures were suggested to arrange more clearly to make the readers to understand the meaning and relationship of these four figures.

4.There are a great deal of works about orientation of hard-brittle crystals available in the literatures. The authors should refer to some recent studies to provide more in-depth discussion. For example, a recent two manuscripts entitled "Crystallographic orientation effect on the polishing behavior of LiTaO3 single crystal and its correlation with strain rate sensitivity, Ceramics International, 2021. In press. https://doi.org/10.1016/j.ceramint.2021.11.324" and "H. Chen, Q. Xu, J. Wang, P. Li, J. Yuan, B. Lyu, J. Wang, K. Tokunaga, G. Yao, L. Luo, Y. Wu, Effect of surface quality on hydrogen/helium irradiation behavior in tungsten, Nuclear Engineering and Technology (2022), doi: https://doi.org/10.1016/j.net.2021.12.006" have provided good guidelines in terms of fracture mode for hard-brittle material and effect of orientation on removal modes.

5.The amount of experimental data in the paper is too small, and the paper mainly focuses on the observation results of equipment. It can be suggested to enrich the experimental process and quantitative data.

6.There are problems with the serial number of the title of the paper, which need to be modified.

Author Response

От имени авторов благодарим за проявленный интерес и проделанную работу. Вы помогли улучшить нашу работу.

Round 2

Reviewer 2 Report

The authors have done significative changes to the manuscript. This contribution should be accepted.

Queries:

(1) Please include the legends (a) and (b) in the images of the Figure 4 to identify them.

Author Response

От коллектива авторов спасибо за проделанную работу!

Включите легенды (a) и (b) в изображения на Рисунке 4, чтобы идентифицировать их.
